# Plant secondary metabolite-dependent plant-soil feedbacks can improve crop yield in the field

**Valentin Gfeller[1], Jan Waelchli[2], Stephanie Pfister[3], Gabriel Deslandes-Hérold[1], Fabio Mascher[4], Gaetan Glauser[5], Yvo Aeby[6], Adrien Mestrot[3], Christelle AM Robert[1], Klaus Schlaeppi[1,2]\*, Matthias Erb[1]\***

[1]Institute of Plant Sciences, University of Bern, Bern, Switzerland; [2]Department of Environmental Sciences, University of Basel, Basel, Switzerland; [3]Institute of Geography, University of Bern, Bern, Switzerland; [4]Department of Plant Breeding, Agroscope, Nyon, Switzerland; [5]Platform of Analytical Chemistry, Université de Neuchâtel, Neuchâtel, Switzerland; [6]Research contracts animals group, Agroscope, Posieux, Switzerland

**\*For correspondence:**
klaus.schlaeppi@unibas.ch (KS);
matthias.erb@ips.unibe.ch (ME)

**Competing interest:** The authors declare that no competing interests exist.

**Abstract** Plant secondary metabolites that are released into the rhizosphere alter biotic and abiotic soil properties, which in turn affect the performance of other plants. How this type of plant-soil feedback affects agricultural productivity and food quality in the field in the context of crop rotations is unknown. Here, we assessed the performance, yield and food quality of three winter wheat varieties growing in field plots whose soils had been conditioned by either wild type or benzoxazinoid-deficient *bx1* maize mutant plants. Following maize cultivation, we detected benzoxazinoid-dependent chemical and microbial fingerprints in the soil. The benzoxazinoid finger-print was still visible during wheat growth, but the microbial fingerprint was no longer detected. Wheat emergence, tillering, growth, and biomass increased in wild type conditioned soils compared to *bx1* mutant conditioned soils. Weed cover was similar between soil conditioning treatments, but insect herbivore abundance decreased in benzoxazinoid-conditioned soils. Wheat yield was increased by over 4% without a reduction in grain quality in benzoxazinoid-conditioned soils. This improvement was directly associated with increased germination and tillering. Taken together, our experiments provide evidence that soil conditioning by plant secondary metabolite producing plants can increase yield via plant-soil feedbacks under agronomically realistic conditions. If this phenomenon holds true across different soils and environments, optimizing root exudation chemistry could be a powerful, genetically tractable strategy to enhance crop yields without additional inputs.

## Editor's evaluation

This study presents findings that are important for understanding plant-soil feedbacks in agriculture. The authors use a large-scale agricultural field experiment to demonstrate the role of root-emitted secondary metabolites in enhancing the yield of the next crop. By using a benzoxazinoid-deficient maize genotype, the authors provide compelling evidence that biomass production and grain yield of several wheat varieties can be increased when grown in soil conditioned by maize plants able to release benzoxazinoids.

## Introduction

Plants alter the soil they live in, and thereby modulate the growth and defense status of other plants (*Bever et al., 1997*). These so-called plant-soil feedbacks can influence plant community composition and ecosystem functions (*Bennett et al., 2017*; *Teste et al., 2017*; *Mariotte et al., 2018*). They have

also been used for centuries in crop rotation schemes to reduce pest, weed and disease pressure and ultimately improve crop yields (*White, 1970*; *van der Putten et al., 2013*). So far, however, proximate mechanistic work on plant-soil feedbacks has rarely been applied to improve crop rotations. Thus, the benefits of this research for the engineering of crop rotations for ecological and sustainable agriculture remain limited (*Mariotte et al., 2018*).

Plant-soil feedbacks can act on a variety of plant traits. Reductions in germination through allelochemicals for instance are common and can be attributed to either direct toxicity or delayed germination resulting from biochemical recognition by the seeds (*Tawaha and Turk, 2003*; *Renne et al., 2014*). Changes in defense expression leading to differences in herbivore performance and preference (*Pineda et al., 2010*; *Kos et al., 2015*; *Hu et al., 2018b*; *Pineda et al., 2020*) and alterations in susceptibility to soil pathogens have also been observed (*Ma et al., 2017*). Finally, changes in hormonal balance have been linked to effects on plant growth and biomass accumulation (*Pieterse et al., 2014*; *Hu et al., 2018a*). The response to plant-soil feedbacks can strongly depend on environmental conditions (*Smith-Ramesh et al., 2017*), and is often species- and variety-specific, thus requiring detailed investigations of defined plant genotypes under realistic environmental conditions (*van der Putten et al., 2013*; *Wagg et al., 2015*; *Hu et al., 2018b*; *Cadot et al., 2021a*). The diversity of plant traits that can be affected call for broad phenotyping efforts that take into account ecologically and economically relevant parameters.

Plants can influence different soil parameters which may then lead to feedback effects. This includes nutrient availability and chemical soil properties (*Bennett and Klironomos, 2019*; *Schandry and Becker, 2020*). Positive feedbacks in agriculture are often attributed to increased soil fertility, water retention, and improved pest control (*Bennett et al., 2012*; *Tamburini et al., 2020*). In recent years, changes in root microbial communities have received substantial attention as drivers of plant-soil feedbacks (*Bever et al., 2012*; *Benitez et al., 2021*). Various plant health benefits have been associated to the rhizosphere microbiome (*Berendsen et al., 2012*) and plant-soil feedbacks represent a promising way to harness these positive effects, including growth promotion and insect resistance in agricultural settings (*Hu et al., 2018b*; *Pineda et al., 2020*).

How do plants alter soil microbial communities? Although multiple mechanisms are likely at play, the release of small molecular weight compounds, including primary and secondary metabolites, is emerging as a major determinant of microbial community composition in the rhizosphere (*Pang et al., 2021*). Flavones, coumarins, triterpenes, and benzoxazinoid secondary metabolites are known to structure the rhizosphere microbiota (*Hu et al., 2018b*; *Stringlis et al., 2018*; *Huang et al., 2019*; *Voges et al., 2019*; *Yu et al., 2021*). Flavones and benzoxazinoids have recently been shown to modulate plant-soil feedbacks via changes in microbial communities (*Hu et al., 2018b*; *Yu et al., 2021*). If and how secondary metabolites can alter plant performance via plant-soil feedbacks under realistic field conditions, however, remains unclear.

Benzoxazinoids are a class of indole-derived secondary metabolites that are produced and released in high quantities by important food crops such as maize and wheat (*Frey et al., 2009*; *Hu et al., 2018b*). Multiple functions of benzoxazinoids have been described, ranging from defense to nutrient uptake (*Niemeyer, 2009*). Soil conditioning by benzoxazinoids can feed back on growth and defense of maize and wheat, where the strength and direction of the feedback depend on the plant genotype and soil characteristics (*Hu et al., 2018b*; *Cadot et al., 2021a*). Benzoxazinoids can shape root microbial communities (*Hu et al., 2018b*; *Cotton et al., 2019*; *Kudjordjie et al., 2019*; *Cadot et al., 2021b*), chelate iron in the soil and possibly reduce the performance of non-benzoxazinoid producing plants via allelopathic effects (*Bigler et al., 1996*; *Niemeyer, 2009*; *Hu et al., 2018a*). Thus, they may influence crop rotations and yields through a variety of plant-soil feedback mechanisms.

To test whether benzoxazinoids can influence crop yields in a rotation scheme, we investigated how maize benzoxazinoids affect weed and pest pressure, germination, growth, yield, and food quality of wheat plants in the field. Maize and wheat are among the most important crops in global food production and are commonly cultivated in sequence in rotation schemes. In a 2-year field experiment involving wild type and benzoxazinoid-deficient *bx1* mutant maize plants, we first evaluated the effects of benzoxazinoid soil conditioning on soil chemistry and microbial communities. In the following season, we planted three wheat varieties into the same field and quantified a wide variety of agronomically important traits, including growth, weed cover, insect damage, yield and yield quality.

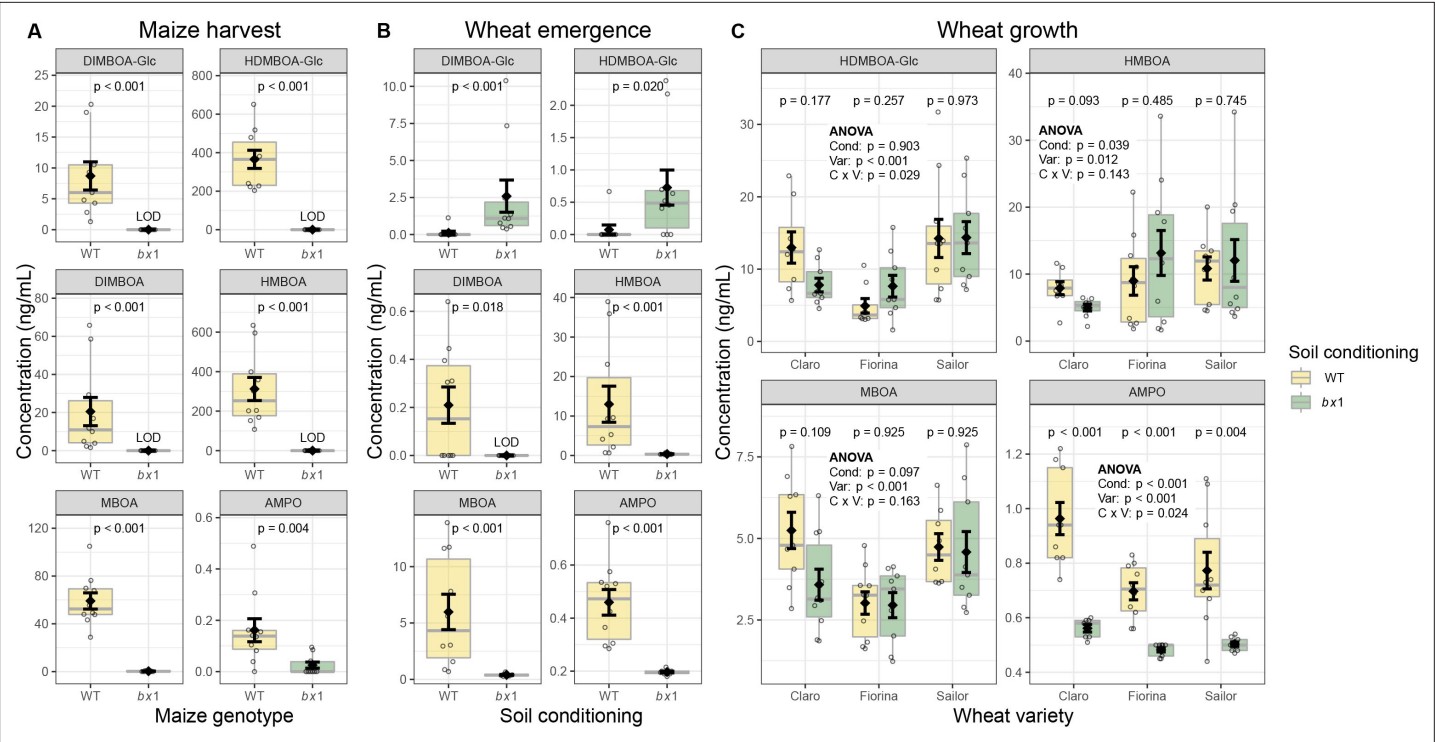

**Figure 1.** Benzoxazinoid soil conditioning results in persistent chemical fingerprints. (**A**) Concentrations of benzoxazinoids in field soil at harvest of wild type (WT) or benzoxazinoid-deficient *bx1* mutant maize plants. (**B**) Benzoxazinoids in field soil 6 weeks after maize harvest. For (**A**) and (**B**) means ±SE, boxplots, and individual datapoints are shown and Wilcoxon rank-sum tests are included (FDR-corrected *p* values, n=10). (**C**) Benzoxazinoids in field soil at wheat growth. Means ±SE, boxplots, and individual datapoints are shown (n=10). ANOVA tables and pairwise comparisons within each wheat variety are included (FDR-corrected p values). LOD: below limit of detection. Cond: soil conditioning (WT or *bx1*). Var: wheat variety. 'C x V': interaction between conditioning and wheat variety. Note that in (**C**) the minimum values of the y-axes were set to a value greater than zero for clearer visualization of treatment differences.

The online version of this article includes the following source data and figure supplement(s) for figure 1:

**Source data 1.** Data of benzoxazinoid concentrations shown in *Figure 1* and of maize shoot dry weight and soil nutrient levels at maize harvest shown in *Figure 1—figure supplement 2*.

**Figure supplement 1.** Experimental set-up.

**Figure supplement 2.** Additional parameters maize harvest.

Through this experiment, we demonstrate that root exudation of secondary metabolites can be linked directly to improved food production under an agronomically realistic crop rotation scenario.

## Results

### Benzoxazinoid soil conditioning results in persistent chemical fingerprints

To test the hypothesis that maize benzoxazinoids modulate the performance of wheat in a crop rotation scheme, we grew wild type W22 and benzoxazinoid-deficient *bx1* mutant maize plants (in a W22 background) in the field. Compared to its wild type counterpart, the *bx1* mutant exhibits a strong reduction in benzoxazinoid production due to a transposon insertion in the *Bx1* gene (*Tzin et al., 2015*). Previous work has shown that consistent soil conditioning and feedback effects can be triggered by different *bx1* mutant alleles in different genetic backgrounds (*Hu et al., 2018b*). Wild type and *bx1* mutant plants were sown separately in 10 strips. The strips themselves were arranged in an alternating pattern, with a strip containing wild type plants followed by a strip containing *bx1* mutant plants, and so on (*Figure 1—figure supplement 1*). Each strip consisted of 12 rows of maize of one genotype. Both maize genotypes grew similarly and accumulated the same amount of biomass at the end of the growing season, most likely due to abundant micronutrients and low pest pressure (*Figure 1—figure*

*supplement 2A*). Substantial amounts of benzoxazinoids and benzoxazinoid degradation products were detected in the soils of plots cultivated with wild-type plants (*Figure 1A*). HDMBOA-Glc was the most abundant benzoxazinoid, followed by HMBOA, DIMBOA, and DIMBOA-Glc. The breakdown products MBOA and AMPO were also detected. Most benzoxazinoids were below the limit of detection in the soils planted with *bx1* mutant plants, indicating that concentrations of the highly emitted compounds (e.g. HDMBOA-Glc) were more than 100 times lower in *bx1* mutant compared to wild type conditioned soil.

To evaluate the persistence of this chemical fingerprint at the time of cultivation of the next crop, we determined benzoxazinoid profiles 6 weeks after maize harvest, at the beginning of winter wheat cultivation. Most benzoxazinoids were 3- to 800-fold less abundant than at the end of maize cultivation (*Figure 1B*). We therefore concentrated the samples prior to analysis, resulting in the detection of traces of benzoxazinoids also in *bx1* mutant conditioned soils. Concentrations of the stable breakdown product AMPO increased more than twofold. DIMBOA, HMBOA, MBOA and AMPO were present in higher concentrations in wild type conditioned soils. Only trace levels of the two glycosylated benzoxazinoids, DIMBOA-Glc and HDMBOA-Glc were found, and their concentrations were higher in *bx1* mutant conditioned soils. As benzoxazinoids are released as glycosides and deglycosylated in the soil, this result is indicative of faster deglycosylation in wild type conditioned soils.

To test if soil conditioning by benzoxazinoids also affected other soil edaphic factors, we analyzed soil macro and micronutrient levels and pH at the end of maize cultivation. No significant differences were found between soils cultivated with wild type or *bx1* mutant plants (*Figure 1—figure supplement 2B*).

We then sowed two different wheat varieties (Claro and Fiorina) into the field, forming strips that went across the different maize strips, resulting in a checkerboard pattern with 20 plots per wheat variety with a size of 6 * 6 m where half of the plots were previously cultivated with wild type and the other half with *bx1* mutant maize (*Figure 1—figure supplement 1*). An additional variety (Sailor) was sown for seed multiplication adjacent to the two other varieties on the same field by a seed company. While Claro and Fiorina were managed without plant protection products, Sailor was treated with herbicides. As Sailor was sown within the premises of our conditioning experiment, we took the opportunity to also measure a subset of performance traits in this variety.

To test if the chemical fingerprint persisted further as wheat grew in soil, we analyzed the soil benzoxazinoids again during wheat growth. As benzoxazinoids are also produced by wheat, the measurements likely represent both old maize and newly wheat produced metabolites. The previous differences in benzoxazinoid levels of HDMBOA-Glc, HMBOA, and MBOA were not detected any more at this point, and DIMBOA-Glc was no longer detected at all (*Figure 1C*). However, AMPO levels were still significantly and consistently higher in plots conditioned with wild type maize across all three wheat varieties. Taken together, these results show that modulating benzoxazinoid production results in a persistent soil chemical fingerprint that is still present in the next crop generation, even if the next crop also produces benzoxazinoids.

## Benzoxazinoid soil conditioning transiently structures rhizosphere microbial communities

To investigate if differences in benzoxazinoid soil conditioning affected the bacterial and fungal communities, we analyzed soil, rhizosphere, and root samples by profiling the bacterial 16 S rRNA gene and the ITS1 region of the ribosomal operon for fungi. At maize harvest, the biggest taxonomic differences at the phylum level were found between the three compartments: root, rhizosphere, and soil (*Figure 2—figure supplement 1*). Permutational Multivariate Analysis of Variance (PERMANOVA) on Bray-Curtis distances revealed significant differences between genotypes in bacterial and fungal community composition of roots and rhizosphere after taking the effect of the sequencing run into account (*Figure 2A*). 9.7 % to 15.7 % of the total variation within compartments was explained by the maize genotype. The benzoxazinoid effect on bacterial and fungal communities was comparable in the roots ($R^2$ bacteria = 13 %, $R^2$ fungi = 12.7 %). In the rhizosphere, we found a more pronounced effect on the fungal community relative to the bacterial community (bacteria = 9.7 %, fungi = 15.7 %). In bulk soil, no benzoxazinoid effects were detected (*Figure 2A*). In line with PERMANOVA, visualization of bacterial and fungal communities by Constrained Analysis of Principal Coordinates (CAP) showed a clear differentiation between maize genotypes in the roots and rhizosphere (*Figure 2B*).

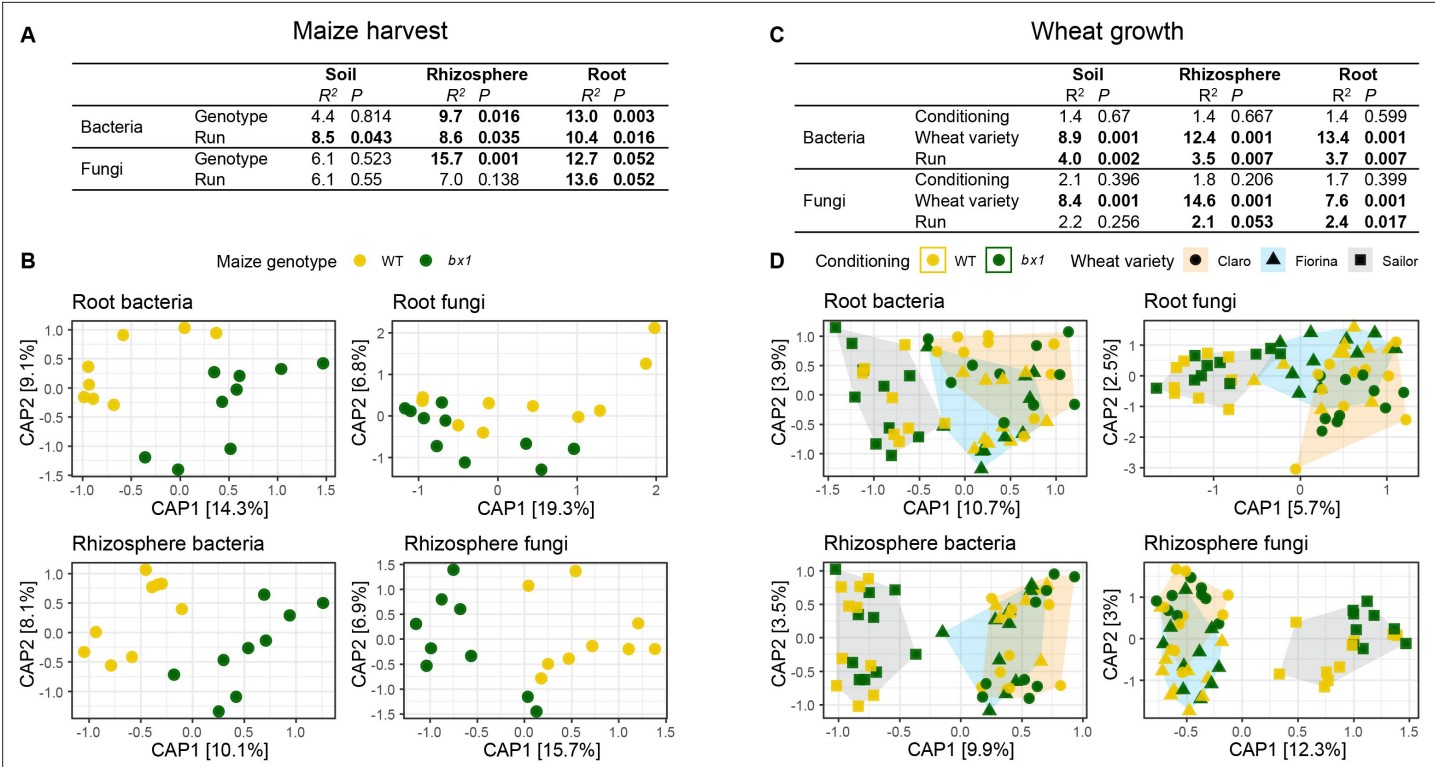

**Figure 2.** Benzoxazinoid soil conditioning transiently structures rhizosphere microbial communities. Soil, rhizosphere, and root-associated microbial communities at maize harvest (**A, B**) and during wheat growth (**C, D**). (**A**) Output of PERMANOVA on Bray-Curtis dissimilarities of bacteria and fungi showing R² and p values for genotype and sequencing run effects in soil, rhizosphere, and root compartments. Significant effects are indicated in bold. (**B**) Constrained Analysis of Principal Coordinates (CAP) confirming the genotype effects found in the PERMANOVA, axis labels denote percentage of explained variance (n=8–10). (**C, D**) Same as in (**A, B**) but also including the factor wheat variety (n=6–10).

The online version of this article includes the following figure supplement(s) for figure 2:

**Figure supplement 1.** Relative abundance of microbial phyla at maize harvest.

**Figure supplement 2.** Relative abundance of microbial phyla in the wheat feedback phase.

Overall, these results confirm that benzoxazinoids structure root-associated microbial communities in maize.

To test if the benzoxazinoid effects on microbial community composition persists, we analyzed bacteria and fungi in the root, rhizosphere, and soil compartments during wheat maturation. Again, the strongest taxonomic differences at phylum level were found between compartments (*Figure 2— figure supplement 2*). PERMANOVA revealed a consistent difference in community composition between wheat varieties, with Sailor being the most dissimilar to the others (*Figure 2C/D*). Note that these differences could also be the result of different positions in the field or crop management of Sailor (*Figure 1—figure supplement 1*). PERMANOVA did not reveal any benzoxazinoid-dependent effects on microbial community composition. Thus, there was no clear legacy effect on microbial community composition at the onset of wheat maturation.

## Benzoxazinoid soil conditioning improves wheat emergence and growth

To investigate whether benzoxazinoid soil conditioning affects wheat performance, we measured emergence shortly after sowing as well as leaf chlorophyll content, plant height and aboveground biomass during wheat growth of the three varieties. Overall, wheat seedling emergence was increased by 8% in benzoxazinoid conditioned soils (*Figure 3A*). Chlorophyll content measured in the youngest fully developed leaf (as a proxy for early plant performance) was also increased in plants growing in benzoxazinoid conditioned soils (*Figure 3B*). Wheat growth, height, and biomass production

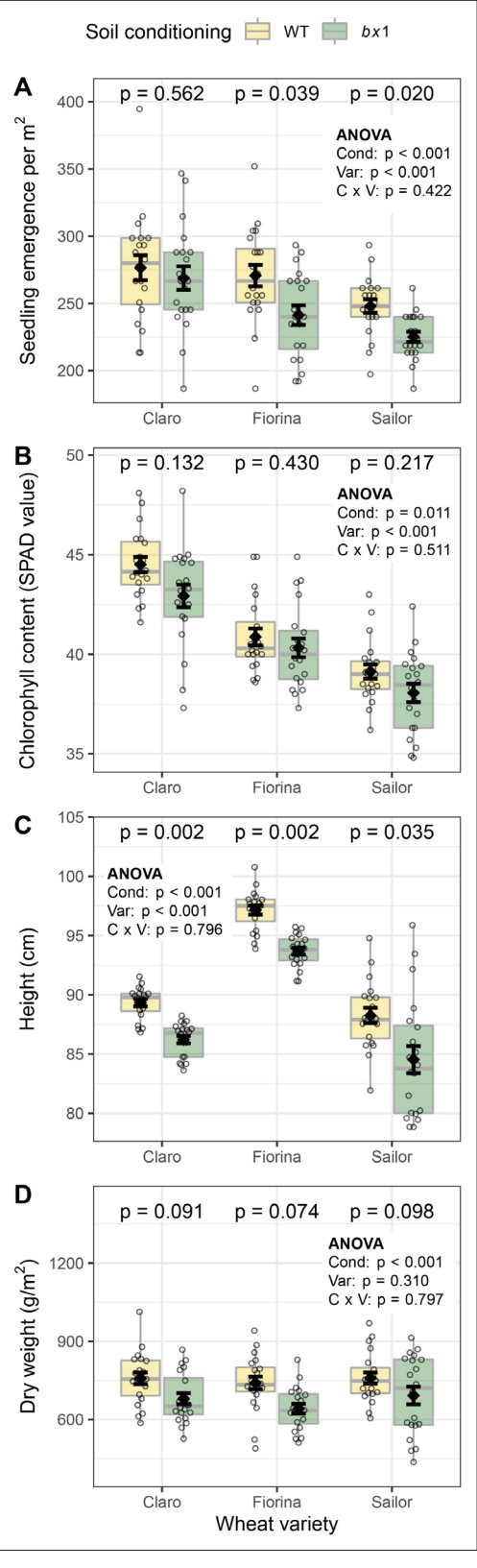

**Figure 3.** Benzoxazinoid soil conditioning improves wheat emergence and growth. (**A**) Seedling emergence, (**B**) chlorophyll content, (**C**) plant height, and (**D**) shoot dry weight of three wheat varieties sown

*Figure 3 continued on next page*

*Figure 3 continued*

in soils previously conditioned with wild type (WT) or benzoxazinoid-deficient *bx1* mutant maize. Means ±SE, boxplots, and individual datapoints are shown (n=20). ANOVA tables and pairwise comparisons within each wheat variety (FDR-corrected *p* values) are included. Cond: soil conditioning (WT or *bx1*). Var: wheat variety. 'C x V': interaction between conditioning and wheat variety. Note that the minimum values of the y-axes were set to a value greater than zero for clearer visualization of treatment differences.

The online version of this article includes the following source data and figure supplement(s) for figure 3:

**Source data 1.** Data of phenotypes shown in *Figure 3* and *Figure 3—figure supplement 1*.

**Figure supplement 1.** Additional benzoxazinoid soil conditioning effects on wheat growth.

per area, as well as shoot water content were also enhanced (*Figure 3C/D*, *Figure 3—figure supplement 1*). Thus, benzoxazinoid soil conditioning increases wheat performance across different wheat varieties.

## Benzoxazinoid soil conditioning does not change weed pressure, but reduces insect infestation

To test for possible changes in weed pressure due to allelopathic effects, we surveyed the weed cover on all plots. Chickweed (*Stellaria media*), Persian speedwell (*Veronica persica*), and Shepherd's purse (*Capsella bursa-pastoris*) were the most abundant weeds. We found that weed pressure differed along the field, and therefore accounted for positional effects in the analysis. We found no effect of soil conditioning status on weed abundance for the varieties Claro and Fiorina (*Figure 4A*). No weeds were detected with the variety Sailor, as this variety was treated with herbicides.

The main herbivore that occurred in the field was the cereal leaf beetle *Oulema melanopus*. The abundance of *Oulema* larvae was significantly reduced on wheat plants of all three varieties grown in benzoxazinoid conditioned soils, with the biggest difference in the variety Sailor (*Figure 4B*). To investigate whether this pattern resulted in reduced damage, we quantified the consumed leaf area on the flag leaves at the end of *Oulema* development. No clear effect was found in terms of insect damage, apart from a tendency of decreased damage on Sailor (*Figure 4C*). We also measured defense

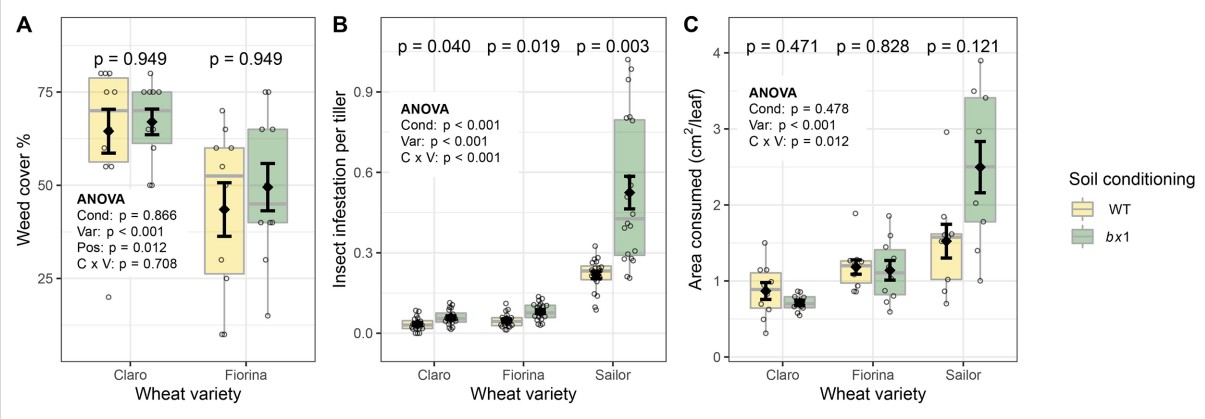

**Figure 4.** Benzoxazinoid soil conditioning does not change weed pressure, but reduces insect infestation. (**A**) Total ground cover by weed plants in plots of three wheat varieties growing in soils previously conditioned with wild type (WT) or benzoxazinoid-deficient *bx1* mutant maize (n=10). No weeds were detected in plots with the variety Sailor due to herbicide treatment of this variety. (**B**) Mean abundance of cereal leaf beetles (*Oulema melanopus*) per tillers (n=20) and (**C**) Consumed flag leaf area by cereal leaf beetles (n=9–10). Means ±SE, boxplots and individual datapoints (n=20) are shown. ANOVA tables and pairwise comparisons within each wheat variety (FDR-corrected *p* values) are included. Cond: soil conditioning (WT or *bx1*). Var: wheat variety. 'C x V': interaction between conditioning and wheat variety. Pos: position on the field. Note that in (**A**) the minimum value of the y-axis was set to a value greater than zero for clearer visualization of treatment differences.

The online version of this article includes the following source data and figure supplement(s) for figure 4:

**Source data 1.** Data of phenotypes shown in *Figure 4* and of leaf phytohormone levels shown in *Figure 4—figure supplement 1*.

**Figure supplement 1.** Benzoxazinoid soil conditioning does not affect leaf phytohormone levels of wheat.

hormone levels, indicative for defense activation. No significant influence of benzoxazinoid soil conditioning was found (*Figure 4—figure supplement 1*).

## Benzoxazinoid soil conditioning increases wheat biomass and yield without compromising grain quality

To understand how benzoxazinoid soil conditioning influences mature wheat plants, we also quantified plant performance at harvest. All wheat varieties had a higher number of tillers per area in benzoxazinoid conditioned soils (*Figure 5A*). To test if these differences can be attributed to differences in emergence or differences in tillering, we also counted the number of tillers per plant. Overall, plants in benzoxazinoid conditioned soils produced a higher numbers of tillers per plant. This pattern was consistent across all varieties, with the most pronounced difference in the variety Sailor (*Figure 5B*).

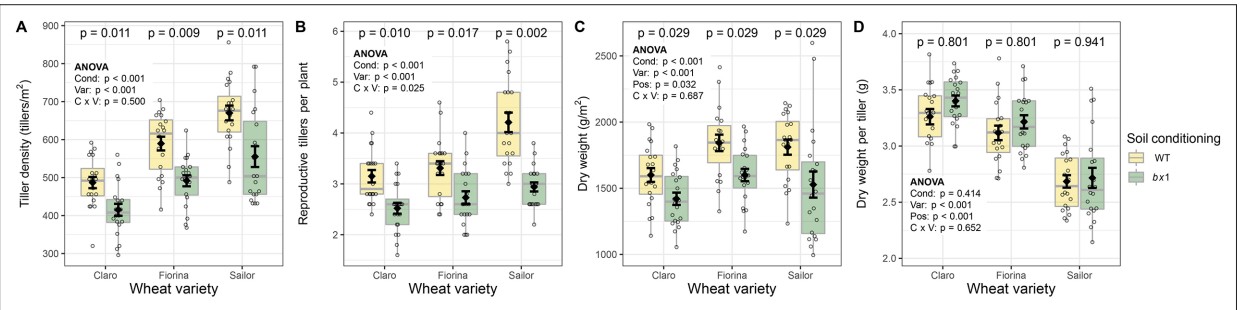

**Figure 5.** Benzoxazinoid soil conditioning increases wheat density and biomass. (**A**) Tiller density, (**B**) reproductive tillers per plant, (**C**) shoot dry weight, and (**D**) dry weight per tiller of the three wheat varieties growing in soils previously conditioned with wild type (WT) or benzoxazinoid-deficient *bx1* mutant maize. Means ±SE, boxplots, and individual datapoints (n=20) are shown. ANOVA tables and pairwise comparisons within each wheat variety (FDR-corrected *p* values) are included. Cond: soil conditioning (WT or *bx1*). Var: wheat variety. 'C x V': interaction between conditioning and wheat variety. Pos: position on the field. Note that the minimum values of the y-axes were set to a value greater than zero for clearer visualization of treatment differences.

The online version of this article includes the following source data for figure 5:

**Source data 1.** Data of phenotypes shown in *Figure 5*.

Next, we measured if the higher tiller density resulted in a higher aboveground biomass per area. Consistent with the results during wheat growth, benzoxazinoid soil conditioning also increased biomass at plant maturity (*Figure 5C*). The weight of individual tillers was similar (*Figure 5D*), demonstrating that benzoxazinoid soil conditioning increased biomass by promoting tiller density, both through enhanced germination and tillering, rather than tiller weight.

We evaluated whether benzoxazinoid soil conditioning improved wheat yield and quantified kernel weight per plot at harvest. For each Claro and Fiorina plot, 9 m$^2$ were harvested. Final yield could not be determined for Sailor, as this field was harvested in bulk for seed multiplication. Yield in both Claro and Fiorina was increased by 4–5% on benzoxazinoid conditioned soils (*Figure 6A*); the productivity of both varieties fell within the expected range of top-yielding winter wheat varieties in Switzerland (variety testing Agroscope). The number of kernels per tiller, the kernel weight per tiller and the thousand kernel weight did not differ between soil conditioning treatments (*Figure 6B*, *Figure 6—figure supplement 1A, B*), showing that the increase in yield is primarily the result of more kernels being produced per area.

To investigate whether the increased wheat yield comes with a penalty in terms of grain quality, we first determined a number of physical kernel properties. Volume per weight, kernel surface area, kernel length and kernel width were not affected by soil conditioning (*Figure 6—figure supplement 1C–F*). We further assessed various agronomically important parameters that are indicative of kernel quality and suitability for baking. We measured protein content, Zeleny index, falling number, as well as dough water absorption, stability and softening. Kernel quality and baking quality were high and showed no differences between soil conditioning treatments (*Figure 6C–E*, *Figure 6—figure supplement 1G–I*). To test if micronutrient content is affected by soil conditioning, we also quantified 21 elements in the harvested wheat kernels. No benzoxazinoid conditioning effects were found (*Figure 6F*, *Figure 6—figure supplement 2*). Taken together, these results demonstrate that maize benzoxazinoid soil conditioning increases wheat yield without affecting kernel quality.

## Discussion

Plants exude secondary metabolites into the rhizosphere and thereby influence the growth and defense of subsequently growing plants (*Hu et al., 2018b*; *Cadot et al., 2021a*). Whether this phenomenon is also relevant in the field, and whether it can be exploited to improve crop productivity, is unknown. Here, we demonstrate that maize plants releasing benzoxazinoids can improve plant growth and crop yield via plant-soil feedbacks under agronomically realistic conditions. Below, we discuss the mechanisms underlying this phenomenon as well as its potential to improve sustainable food production.

Translating plant-soil feedback mechanisms to crop resistance and productivity has been proposed as a promising approach in sustainable agriculture (*Mariotte et al., 2018*). Plant secondary metabolites and their degradation products are known to suppress the growth of other plants (*Schandry and Becker, 2020*) and improve herbivore and pathogen resistance (*Niemeyer, 2009*). Less is known about their potential to influence seedling establishment (*Lamichhane et al., 2018*), yield quantity and yield quality in the field (*Cadot et al., 2021a*; *Pang et al., 2021*). We found that benzoxazinoid soil conditioning by the preceding crop increased subsequent wheat emergence, tillering and plant performance in the field, resulting in higher plant biomass and kernel yield. Because weed cover was unaffected, and increased insect infestation did not result in increased leave damage, we conclude that the positive effects on yield were the result of directly improved germination and tillering rather than changes in plant competition or pest damage. Interestingly, the observed increase in biomass is different from what was observed in an earlier greenhouse study (*Cadot et al., 2021a*). This discrepancy is partially explained by the fact that the greenhouse study investigated individual plant performance and did thus not take into account germination effects. Clearly, agriculturally relevant field experiments are useful and necessary to quantify the costs and benefits of plant-soil feedbacks for sustainable agriculture.

In crop rotations the identity of the preceding crop is known to affect growth, tiller density, yield, and kernel protein content of wheat (*Anderson, 2008*; *Rieger et al., 2008*; *Sieling and Christen, 2015*). Our work expands this knowledge by demonstrating that the release of chemicals by the preceding crop is sufficient to enhance overall crop yield through enhanced germination and tillering. Although higher plant densities are often associated with lower grain quality (*Bastos et al., 2020*), we found that the yield increase did not affect physical parameters, grain micronutrient composition,

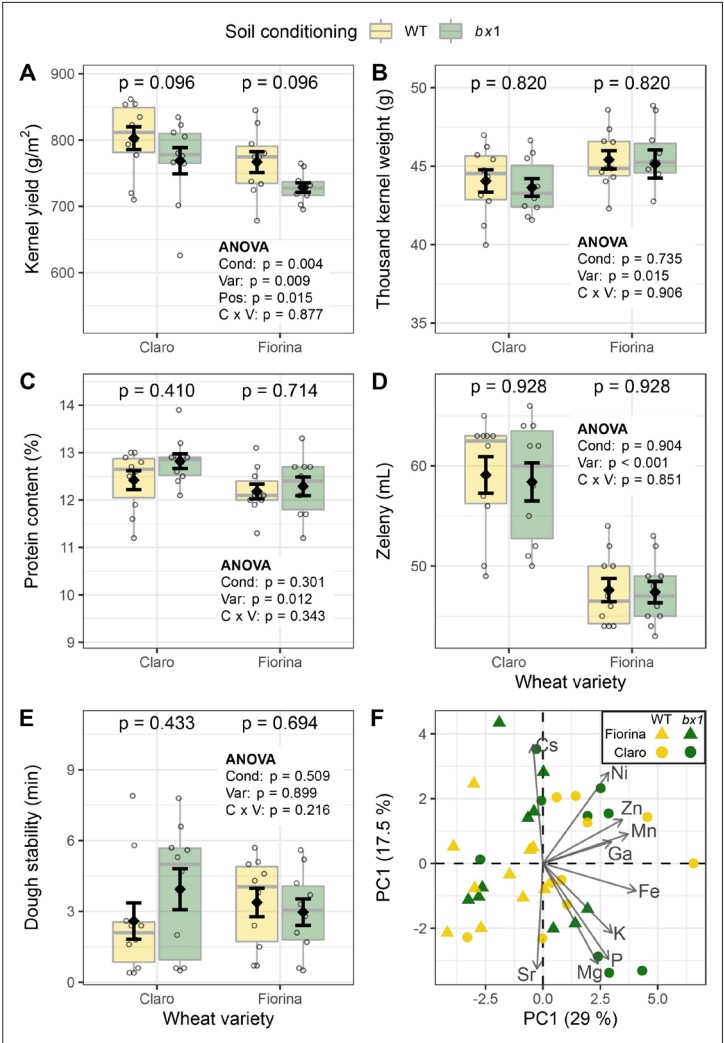

**Figure 6.** Benzoxazinoid soil conditioning increases wheat yield without compromising grain quality. (**A**) Yield of two wheat varieties growing in soils previously conditioned with wild type (WT) or benzoxazinoid-deficient *bx1* mutant maize. Kernel quality measures included (**B**) thousand kernel weight, (**C**) kernel protein content, (**D**) Zeleny index (flour quality), (**E**) dough stability, and (**F**) PCA of kernel micronutrient composition. For (**A–E**) means ±SE, boxplots, and individual datapoints are shown (n=10). ANOVA tables and pairwise comparisons within each wheat variety (FDR-corrected p values) are included. (**F**) reports the first two axes of the micronutrient PCA, including individual samples and the contribution of the 10 elements explaining most of the variation in the dataset (arrow length denotes relative contribution). Cond: soil conditioning (WT or *bx1*). Var: wheat variety. 'C x V': interaction between conditioning and wheat variety. Pos: position on the field. Note that the minimum values of the y-axes were set to a value greater than zero for clearer visualization of treatment differences.

The online version of this article includes the following source data and figure supplement(s) for figure 6:

**Source data 1.** Data of phenotypes shown in *Figure 6*, *Figure 6—figure supplement 1* and *Figure 6—figure supplement 2*.

**Figure supplement 1.** Benzoxazinoid soil conditioning does not affect wheat kernel measurements and baking quality.

**Figure supplement 2.** Benzoxazinoid soil conditioning does not affect micronutrient concentrations in wheat kernels.

grain quality, and baking quality. The increase in yield of 4–5% is equivalent to more than two years of breeding (*Le Gouis et al., 2020*), and represents a true advantage because quality remained constant without additional agricultural inputs. Benzoxazinoid exudation and responsiveness to benzoxazinoid soil conditioning are thus promising targets for future breeding efforts. Future crop rotations could be

designed using varieties that are optimized for such traits. One can for instance envisage a scenario where high benzoxazinoid maize hybrids are selected specifically to precede highly responsive wheat cultivars. Future field experiments will have to evaluate how other crops respond to benzoxazinoid conditioning in the field and how generalizable the obtained results are across different years and locations. Furthermore, a better understanding of the genetic basis of benzoxazinoid exudation will be helpful to boost the release of benzoxazinoids into the soil. Such work will help to further unlock the potential of plant-soil feedbacks for the much needed sustainable intensification of agriculture (*Hunter et al., 2017*; *Mariotte et al., 2018*).

Plant-soil feedbacks can be triggered by different mechanisms (*Bennett and Klironomos, 2019*): the first plant generation changes soil chemistry (*Schandry and Becker, 2020*), root-associated microbiota (*Bever et al., 2012*) or their interaction, with changes in chemistry mediating changes in microbiota (*Hu et al., 2018b*; *Yu et al., 2021*). The persistence of chemical and microbiological changes is seen as a key factor in this context. It has been proposed that chemical changes may be more short lived than microbial changes, as plant secondary metabolites can be degraded rapidly (*Bennett and Klironomos, 2019*). In line with previous studies, we found benzoxazinoids alter the composition of root-associated microbes (*Hu et al., 2018b*; *Cotton et al., 2019*; *Kudjordjie et al., 2019*; *Cadot et al., 2021b*). However, these effects disappeared by the end of the vegetative growth of the next crop. By contrast, the benzoxazinoid chemical fingerprint persisted across both cultivation periods. AMPO, a microbial degradation product with a half life of months (*Macías et al., 2004*; *Niemeyer, 2009*), was found in higher concentrations in benzoxazinoid conditioned soils of all three wheat varieties. Thus, we conclude that the benzoxazinoid chemical fingerprint is more long-lived that the change in the microbial fingerprint. Microbial community changes may still have contributed to plant-soil feedback effects, as many of the late phenotypes (*Figure 5*, *Figure 6*) were explained by initial differences in germination and tillering, where transient microbiome effects could still have been stronger. More research is needed to disentangle the relative importance of chemical and microbial fingerprints which will aid to optimize the design of agroecologically smart crop rotations.

## Conclusions

This study presents a proof of concept for the utilization of plant root exuded metabolites to increase agricultural yield without additional external inputs. Our findings open new avenues to optimize plant traits in crop rotations for a more sustainable agriculture. Future studies with different varieties and crop species and in a wider range of soils and under various farming regimes will help to unravel the generalizability and applicability of using exudate-mediated plant-soil feedbacks in sustainable agriculture.

## Materials and methods

### Plant material

The field experiment was conducted in two phases, the conditioning phase with maize (*Zea mays*) and the feedback phase with wheat (*Triticum aestivum*, *Figure 1—figure supplement 1*). The wild type maize inbred line W22 and the benzoxazinoid-deficient *bx1* transposon knockout mutant (gene identifier GRMZM2G085381; Ds, B.W06.0775) (*Tzin et al., 2015*) were kindly provided by Georg Jander (Cornell University, Ithaca, NY; USA). The inbred lines were surrounded by a buffer zone of the hybrid maize variety Gottardo. In the feedback phase the wheat varieties CH Claro (referred to as Claro), Fiorina, and Sailor were grown. All three wheat varieties were provided by Saatzucht Düdingen (Düdingen; Switzerland) and are commonly cultivated in Switzerland (recommended varieties by Agroscope). Claro is an obligate winter wheat, Fiorina can be cultivated as winter or spring wheat, and Sailor is a common forage winter wheat variety.

### Experimental setup

The conditioning phase indicates the first season where the field was cultivated with wild type and *bx1* mutant maize to condition the soil with or without benzoxazinoids. 'Benzoxazinoid soil conditioning' refers to the process of benzoxazinoid exudation into the surrounding soil and the resulting changes in the soil (e.g. microbial community composition). In the second season, that is the feedback phase, wheat was grown to survey the effects of previous benzoxazinoid soil conditioning on wheat

performance. To test for genotype-specific responses, we investigated two wheat varieties Claro and Fiorina. In addition, the seed company Saatzucht Düdingen has grown a third wheat variety (Sailor) adjacent to our two wheat varieties, and we were kindly allowed to investigate a subset of pheno-types for that variety as well. Therefore, we had three wheat varieties to survey during growth, but could not obtain data on yield and kernel quality for Sailor (*Figure 1—figure supplement 1*). At the end of the conditioning phase, maize biomass, belowground microbiota and soil parameters, including benzoxazinoids, were measured. In the feedback phase we determined wheat emergence, growth, and weed and insect infestation. Soil benzoxazinoids and microbiota were analyzed again during wheat growth. At the end of the feedback phase kernel quantity and quality were evaluated (*Figure 1—figure supplement 1*). For detailed methods see below.

### Field specifications

The experiment was carried out in 2019 and 2020 on a field at the Agroscope research station in Posieux, Switzerland (parcel 2.3, 46°46′23.09″N 7°06′22.95″E). The soil was classified as a sandy loam. The cropping history of this field was a fodder meadow (mixture of red clover and Italian ryegrass; 2018), winter barley (2017), triticale and alfalfa (field divided, 2016), maize and alfalfa (field divided, 2015), alfalfa and maize (field divided, 2014), and alfalfa (2012–2013). The crops were managed according to Swiss conventional agricultural practices by the field team of Agroscope and the educa-tion farm of the Agricultural competence center in Grangeneuve, nearby Posieux. There was a long-lasting drought period in spring 2020 (feedback phase).

### Maize conditioning phase

Wild type and *bx1* inbred lines were alternately sown in 5 strips of 12 rows each (*Figure 1—figure supplement 1*). Distance between maize rows was 75 cm, distance between plants within a row was 15 cm. The inbred lines were surrounded by a minimum of 18 rows of hybrid maize. Before sowing, the soil was fertilized with manure (40 m³/ha), ploughed, and harrowed. Weeds were once treated with herbicide (Equip Power 1.5 l/ha). During plant growth, maize was fertilized twice, firstly with ammonium nitrate supplemented with sulfur 100 kg/ha (25% N, 5% Mg, 8.5% S) and secondly with urea 180 kg/ha (46% N). Maize was harvested and silaged after 22 weeks. One week before harvest, 4 plants per maize strip were randomly selected for phenotyping resulting in 20 replicates per geno-type (wild type and *bx1*). The aboveground biomass was harvested, dried at 80 °C and weighed. For half of the samples (n=10), soil cores of 20 x 20 x 20 cm containing the root system were excavated and used for analysis of benzoxazinoid concentrations, microbiomes, and further soil parameters as described below.

### Overview wheat feedback phase

The wheat varieties were sown one week after maize harvest. Claro and Fiorina were sown in two alternating strips, each perpendicular to the orientation of the maize rows (*Figure 1—figure supple-ment 1*). Sailor was sown in the same orientation as the maize. Distance between wheat rows was 12.5 cm. Prior to sowing the soil was harrowed. During plant growth, wheat was fertilized twice, first with 50 kg N/ha of urea-ammonium nitrate solution (UAN; 39 % N) combined with 120 kg/ha Kiserite (15% Mg, 20% S) and second with 55 kg N/ha of UAN solution (39 % N). No plant protection products were applied to Claro and Fiorina, whereas the field of Sailor was treated with a herbicide against weeds. Four weeks after sowing, at wheat emergence, soil samples were taken for benzoxazinoid analysis. With a soil sampler, 10 soil cores per plot (17 mm diameter, 20 cm deep) were taken and combined to one sample (n=10 per soil conditioning). Germination, plant growth, and insect infesta-tion were phenotyped as described below. During wheat growth, at the end of the vegetative phase, soil cores (7x7 cm wide, 12 cm deep) were taken below three randomly selected wheat plants per plot and pooled for benzoxazinoid and microbiome analysis (n=10 per treatment combination). After 41 weeks of growth, the wheat was harvested (see below).

### Phenotyping feedback phase

To survey benzoxazinoid-dependent plant-soil feedbacks on wheat growth, we measured various parameters. Phenotyping was carried out on all subplots (*Figure 1—figure supplement 1*), resulting in 20 replicates for each combination of soil conditioning status (wild type, *bx1*) and wheat variety

(Claro, Fiorina, Sailor). Weed cover estimation, determination of insect damage, and harvesting was done on plot level, resulting in 10 replicates for each treatment combination.

Emerged seedlings were counted on 1.5 m of a randomly selected wheat row within each subplot one month after the wheat was sown. Seedling emergence per m$^2$ was calculated. At the end of tillering, we measured chlorophyll content with a SPADE-502 chlorophyll meter (Konica Minolta, Tokyo; Japan). Chlorophyll was determined in the middle of the youngest fully expanded leaf of 20 randomly selected plants per subplot and the mean value was recorded. During stem elongation, weed abundance was surveyed by estimating percentage weed cover per plot. At the end of the vegetative growth stage plant height of 10 randomly selected plants per subplot was measured and averaged for analysis. In addition, biomass accumulation was measured, by harvesting wheat plants along 1 m of a randomly selected row per subplot. Fresh biomass was weighed before plant material was dried at 80 °C until constant weight, dry biomass was determined, and plant water content was calculated.

Infestation by the cereal leaf beetle (*Oulema melanopus*) was surveyed at the end of stem elongation. Along 9 m of a row within a subplot all larvae were counted and infestation per m$^2$ was calculated. To determine the total larval damage on the leaves, 10 flag leaves were sampled per plot before the leaves started to wilt. Leaves were transported to the laboratory in a wettened plastic bag stored in a cooled container. Leaves were then scanned and the consumed area per leaf was determined using the R packages *EBImage* and *pliman* (**Pau et al., 2010**; **Olivoto, 2021**). In addition, at the end of the vegetative phase five flag leaves of five plants per plot were randomly selected, wrapped in aluminum foil and snap frozen in liquid nitrogen for later determination of phytohormone levels (see below).

Once the kernels were ripe, total biomass accumulation was determined by harvesting wheat plants along 1 m of a randomly selected row per subplot. Plant material was dried at 80 °C before measuring biomass. To calculate tiller density and weight per tiller, the number of tillers in the dried material were counted. A subsample of five randomly selected heads were threshed with a laboratory thresher (LT-15, Haldrup GmbH, Ilshofen; Germany), and kernels were counted and weighed. Next, we randomly selected five plant per subplot and counted the number of tillers per plant, mean tiller number per plant was taken for statistical analysis.

At the end of the feedback phase, we harvested the experiment plots with a compact plot combine harvester (Zürn 110, Zürn GmbH, Schöntal-Westernhausen; Germany). Yield was determined based on a 9 m$^2$ area in the center of the plots (**Figure 1—figure supplement 1**) and kernel weight per plot was determined. A subset of these kernels was taken for analyzing kernel quality and micronutrient composition (see below).

## Benzoxazinoid analysis

At the end of maize growth, at wheat emergence and during wheat growth soils were sampled as described above and benzoxazinoids and break down products were analyzed. Each soil sample was processed with a test sieve (5 mm mesh size), then 25 mL of soil was transferred into a 50 mL centrifuge tube and homogenized in 25 mL acidified MeOH/H2O (70:30 v/v; 0.1% formic acid). For extraction, the suspension was incubated for 30 minutes at room temperature on a rotary shaker, followed by a centrifugation step (5 min, 2000 *g*) to sediment the soil. The supernatant was passed through a filter paper (Grade 1; Size: 185 mm; Whatman, GE Healthcare Life Sciences, Chicago, IL; United States), 1 mL of the flow through was transferred into a 1.5 mL centrifuge tube, centrifuged (10 min, 19,000 *g*, 4 °C), and the supernatant was sterile filtered (Target2TM, Regenerated Cellulose Syringe Filters. Pore size: 0.45 µm; Thermo Scientific, Waltham, MA; United States) into a HPLC glass tube for further analysis.

To obtain detectable concentrations of benzoxazinoids at wheat emergence and during wheat growth, the samples needed to be concentrated before the second centrifugation step 20 and 10 times, respectively. To obtain that, 20 mL or 10 mL of each sample was completely evaporated at 45 °C (CentriVap, Labconco, Kansas City, MO; USA) and the pellet was resuspended in 1 mL of acidified MeOH/ H2O (70:30 v/v; 0.1% formic acid).

The benzoxazinoid analysis was performed as previously described (**Robert et al., 2017**). Briefly, an Acquity UHPLC system coupled to a G2-XS QTOF mass spectrometer equipped with an electrospray source and piloted by the software MassLynx 4.1 (Waters AG, Baden-Dättwil; Switzerland) was used. Gradient elution was performed on an Acquity BEH C18 column (2.1x50 mm

i.d., 1.7 mm particle size) at 90–70% A over 3 min, 70–60% A over 1 min, 40–100% B over 1 min, holding at 100% B for 2.5 min, holding at 90% A for 1.5 min where A=0.1% formic acid/water and B=0.1% formic acid/acetonitrile. The flow rate was 0.4 mL/min. The temperature of the column was maintained at 40 °C, and the injection volume was 1 µL. The QTOF MS was operated in positive mode. The data were acquired over an m/z range of 50–1200 with scans of 0.15 s at a collision energy of 4 V and 0.2 s with a collision energy ramp from 10 to 40 V. The capillary and cone voltages were set to 2 kV and 20 V, respectively. The source temperature was maintained at 140 °C, the desolvation was 400 °C at 1000 L/hr and cone gas flow was 50 L/hr. Accurate mass measurements (<2 ppm) were obtained by infusing a solution of leucin encephalin at 200 ng/mL and a flow rate of 10 mL/min through the Lock Spray probe. Absolute quantities were determined through standard curves of pure compounds. For that MBOA (6-methoxy-benzoxazolin-2(3 H)-one) was purchased from Sigma-Aldrich Chemie GmbH (Buchs; Switzerland). DIMBOA-Glc (2-O-β-D-glucopyranosyl-2,4-dihydroxy-7-methoxy-2H-1,4-benzoxazin-3(4 H)-one) and HDMBOA-Glc (2-O-β-D-glucopyranosyl-2-hydroxy-4,7-dimethoxy-2H-1,4-benzoxazin-3(4 H)-one) were isolated from maize plants in our laboratory. DIMBOA (2,4-dihydroxy-7-methoxy-2H-1,4-benzoxazin-3(4 H)-one), HMBOA (2-hydroxy-7-methoxy-2H-1,4-benzoxazin-3(4 H)-one), and AMPO (2-amino-7-methoxy-3H-phenoxazin-3-one) were synthesized in our laboratory.

## Soil analysis

A subsample of the soil of each root system excavated at the end of maize growth (see above), was taken and pooled to obtain 4 representative samples of the field per genotype. Soil parameters were then analyzed by LBU Laboratories (Eric Schweizer AG, Thun; Switzerland). Water ($H_2O$), ammonium acetate EDTA (AAE), and carbon dioxide saturated water ($CO_2$) extractions were performed for different nutrients. $H_2O$ extracts serve as a proxy for plant available nutrients, AAE extracts for nutrients available through plant chelation mechanisms and $CO_2$ extracts are a common extraction procedure for magnesium, phosphorus, and potassium (similar to $H_2O$ extracts).

## Phytohormones analysis

Concentrations of salicylic acid (SA), oxophytodienoic acid (OPDA), jasmonic acid (JA), jasmonic acid-isoleucine (JA-Ile) and abscicic acid (ABA) were determined by UHPLC-MS/MS. First, wheat leaf samples were ground to a fine powder under constant cooling with liquid nitrogen. An aliquot of 100 mg (±20%) was taken and the exact weight was noted for the final determination of hormone concentration. Next, phytohormones were extracted as described in *Glauser et al., 2014* with minor adjustments: 10 µL of labelled internal standards (d5-JA, d6-ABA, d6-SA, and 13C6-JA-Ile, 100 ng/mL in water) were added to the samples and hormones were extracted in ethylacetate/formic acid (99.5:0.5, v/v), the samples were centrifuged and evaporated to dryness, and finally resuspended in 200 µL of MeOH 50% for analysis. Two µL of extract were injected in an Acquity UPLC (Waters AG, Baden-Dättwil; Switzerland) coupled to a QTRAP 6500, (Sciex, Framingham, MA; USA). Analyst v.1.7.1 was used to control the instrument and for data processing. Each phytohormone peak was normalized to that of its corresponding labelled form except that of OPDA which was normalized to that of 13C6-JA-Ile.

## Kernel analysis

For morphological analysis of kernels, a subsample of 25 mL of kernels was taken. Volume weight, thousand kernel weight (TKW), kernel surface area, kernel length, and kernel width were determined by means of a microbalance and a MARVIN kernel analyzer (GTA Sensorik GmbH, Neubrandenburg; Germany). A subset of kernels was milled for further analysis. To test flour quality, we determined the falling number (according to ICC standard method 107/1), Zeleny index (according to ICC standard method 116/1) and protein content, which was evaluated by near-infrared reflectance spectroscopy (NIRS) using a NIRFlex N-500 (Büchi Labortechnik AG, Flawil; Switzerland). We further tested dough quality using a micro-doughLAB farinograph (model 1800, Perten Instruments, PerkinElmer, Waltham, MA; USA). Dough stability (min), dough softening (Farinograph Units, FU), and water absorption capacity of the flour (%) during kneading were analyzed according to the manufacturer's protocol.

## Kernel micronutrient analysis

We analyzed total element concentrations for 21 elements as grain micronutrients. Forty g of kernels per plot were ground to fine powder using a cutting mill (Pulverisette, Fritsch, Idar-Oberstein; Germany). Element extraction and analysis was performed as previously described (*Cadot et al., 2021b*), with small adjustments: An aliquot of 250 mg grain powder was extracted in 4 ml of concentrated $HNO_3$ (35%) overnight and 2 mL of $H_2O_2$ (30%) was added. Samples were vortexed for 5 s before microwave extraction at 95 °C for 30 min. Before analysis, tubes were filled to 50 mL with $HNO_3$ (1%) and centrifuged (5 min at 2500 rpm) to remove remaining particles. Elements in the extracts were quantified with inductively coupled plasma mass spectrometry (ICP-MS, 7700 x, Agilent, Santa Clara, CA; USA).

## Microbiota profiling

The sampling of the soil cores on the field was describe above. To prepare the soil samples, the root system was removed from the soil core, subsequently the soil was sieved through a test sieve (mesh size 5 mm). Root and rhizosphere samples were prepared as previously reported (*Hu et al., 2018b*), with minor modifications: Root segments corresponding to –5 to –10 cm below soil level were harvested and large soil particles were removed, before washing the roots twice in a 50 mL centrifuge tube with 25 mL of sterile $ddH_2O$, by vigorously shaking the tube 10 times. The wash fractions were combined, centrifuged (5 min at 3000 *g*) and the resulting pellet was frozen at –80 °C for further processing (rhizosphere sample). The washed roots were freeze-dried for 72 h and subsequently milled to fine powder using a Ball Mill (Retsch GmBH, Haan; Germany) for 30 s at 30 Hz with one 1 cm steel ball.

For DNA extraction, a subsample of 200 mg soil and rhizosphere, and 20 mg of root powder was taken. DNA from all compartments were extracted using the FastDNA SPIN Kit for Soil (MP Biomedicals LLC, Solon, OH; USA) following the manufacturer instruction. In brief, after adding 978 µL of sodium phosphate buffer and 122 µL of MT buffer to each aliquot, the samples were homogenized with a Retsch Mixer Mill during 40 s at 25 Hz. Following 10 min of centrifugation, 250 µL of PPS was added to the supernatant. After mixing ten times by inversion, samples were centrifuged for 5 min. The supernatant was mixed by inversion with 1 mL of binding matrix suspension, transferred to a SPIN filter and then centrifuged for 1 min. The binding matrix was washed with 500 µL of SEWS-M and a total of 3 min of centrifugation was performed. The matrix was air-dried for 5 min, and the binding matrix was resuspended with 100 µL of DNAse/Pyrogen-Free water. After incubating 5 min, DNA was eluted by centrifuging for 1 min. Extraction was performed at room temperature and all centrifugation steps were done with 14,000 *g*. After that step, the DNA was distributed into 96-well plates in a random and equal manner. The DNA concentrations were quantified with the AccuClear Ultra High Sensitivity dsDNA quantification kit (Biotium, Fremont, CA; USA) and diluted to 2 ng µL$^{-1}$ using a Myra Liquid Handler (Bio Molecular Systems, Upper Coomera; Australia).

For the bacterial library, a first PCR reaction was performed with the non-barcoded 16 S rRNA gene primers 799 F (AACMGGATTAGATACCCKG, *Chelius and Triplett, 2001*) and 1193 R (ACGTCATC CCCACCTTCC, *Bodenhausen et al., 2013*). A second PCR tagged the PCR product with custom barcodes. The first PCR program consisted of an initial denaturation step of 2 min at 94 °C, 25 cycles of denaturation at 94 °C for 30 s, annealing at 55 °C for 30 s, elongation at 65 °C for 30 s, and a final elongation at 65 °C for 10 min. The second PCR program was similar, with the difference that the number of cycles was reduced to 10.

For the fungal library, a first PCR reaction was performed with the non-barcoded internal transcribed spacer (ITS) region primers ITS1-F (CTTGGTCATTTAGAGGAAGTAA, *Gardes and Bruns, 1993*) and ITS2 (GCTGCGTTCTTCATCGATGC, *White et al., 1990*). A second PCR tagged the PCR product with custom barcodes. The first PCR program consisted of an initial denaturation step of 2 min at 94 °C, 23 cycles of denaturation at 94 °C for 45 s, annealing at 50 °C for 60 s, elongation at 72 °C for 90 s, and a final elongation at 72 °C for 10 min. The second PCR program was similar, with the difference that the number of cycles was reduced to 7.

All PCR reactions were performed with the 5-Prime Hot Master Mix (Quantabio, QIAGEN, Beverly, MA; USA). All PCR products and pooled library were purified with CleanNGS beads (CleanNA, Waddinxveen; The Netherlands) according to manufacturer protocol with a ratio of 1:1.

All the PCR products were quantified with the AccuClear Ultra High Sensitivity dsDNA quantification kit (Biotium, Fremont, CA; USA) and subpooled by sample type, library type and sequencing run

(*Supplementary file 1*). Subpools were assembled using a Myra Liquid Handler by adding an equal mass of each PCR product. For the bacterial library, the rhizosphere and root subpools were purified on an agarose gel (amplicon ~450 bp) using the NucleoSpin Gel and PCR Clean-up kit (Macherey-Nagel, Düren; Germany), whereas all other subpools were purified with CleanNGS beads. Subpools were quantified with the Qubit dsDNA BR kit (Invitrogen, Thermo Fisher Scientific, Waltham, MA; USA) and equally divided into two sequencing libraries (BE09 & BE10). All samples were paired-end sequenced (v3 chemistry, 300 bp paired end) on an Illumina MiSeq instrument at the NGS platform of the University of Bern (Switzerland).

## Bioinformatics

Raw reads were first quality inspected with *FastQC* and demultiplexed using *cutadapt* (**Andrews, 2010**; **Martin, 2011**). The barcode-to-sample assignments are documented in the **Supplementary file 2**. With *cutadapt* we also removed primer and barcode sequences from the reads (error 0.1, no indels). We utilized the *DADA2* pipeline of **Callahan et al., 2016**; R package *DADA2* to infer exact amplicon sequences variants (ASVs) from the sequencing reads. The raw reads were quality filtered (max. expected errors: 0; max. N's allowed: 0), truncated to the minimal lengths (250 bp, forward read; 170 bp, reverse) and shorter and low quality reads (truncQ = 2) or reads matching PhiX were discarded. The error rates were learned for the separate sequencing runs using the *DADA2* algorithm to denoise the reads and infer true sequence variants. Next, the paired forward and reverse sequences were merged by a minimal overlap of twelve identical bases, a count table was created, and chimeras were removed using the *DADA2* scripts. Finally, the taxonomy was assigned using a *DADA2* formatted versions of the *SILVA* v.132 database (**Quast et al., 2013**; **Callahan, 2018**) for bacteria and the FASTA general release from *UNITE* v8.3 (**Abarenkov et al., 2021**) for fungi.

## Statistical analysis

All statistical analyses were conducted in R (**R Development Core Team, 2021**). Data management and visualization was performed using the *tidyverse* package collection (**Wickham et al., 2019**). Microbiota of root, rhizosphere and soil compartments were analyzed separately for maize samples (conditioning phase) and wheat samples (feedback phase). The variation between sequencing runs was taken into account in all models. We rarefied the data (100 x; depth: bacteria: 8000, fungi: 1'200), because this normalization technique efficiently mitigates artifacts of different sampling depths between sample groups (**Weiss et al., 2017**). Effects on community composition were tested by permutational analysis of variance (PERMANOVA, 999 permutations) on Bray-Curtis distances in the R package *vegan* (**Oksanen et al., 2020**). For maize, we tested for differences between genotypes (model: beta diversity ~genotype + run), and for wheat, we tested for effects of soil conditioning and wheat variety (model: beta diversity ~genotype + variety + run). We visualized the beta diversity by plotting the Canonical Analysis of Principal coordinates (CAP) using the R package *phyloseq* (**McMurdie and Holmes, 2013**).

Plant phenotyping data was analyzed by analysis of variance (ANOVA). Statistical assumptions such as normal distribution and homoscedasticity of error variance were inspected visually from diagnostic quantile-quantile and residual plots. To account for our experimental design and avoid potential pseudo-replication, we included the maize strips as a random blocking factor. To accomplish this, we fitted a linear mixed-effects model using the *lme*() function from the *nlme* package (**Pinheiro et al., 2021**). This function further allowed us to account for unequal variance among treatment groups, if necessary. Possible correlations of the response variables with the position on the field were tested, and, if significant, the position on the field was factored into the model to account for otherwise unexplained variation. For linear models in the feedback phase, we tested for soil conditioning effects within each wheat variety by comparing estimated marginal means (EMMs; *emmeans* package) and reporting false discovery rate (FDR) corrected *p* values (**Benjamini and Hochberg, 1995**; **Lenth, 2022**). Wilcoxon rank-sum tests were performed to test for differences in benzoxazinoid concentrations between wild type and *bx1* conditioned soil in the conditioning phase and at wheat emergence; p values were also FDR adjusted. Maize genotype-dependent differences on soil parameters were tested by Welch's two-sample *t*-test and p values were FDR adjusted. Possible differences in element profile of wheat kernels were visualized through Principal Component Analysis (PCA, *FactoMineR*

package; *Lê et al., 2008*). The 10 elements explaining most of the variance in PCA-axes 1 and 2 were visualized as arrows.

## Acknowledgements

We thank Jean-François Rauber, Wolfram Schuwey and Raphaël Grandgirard (Agricultural competence center canton Freiburg, Grangeneuve, Switzerland) for their field assistance and Lilia Levy, Lydia Michaud and Noemi Schaad (Agroscope, Changins, Switzerland) for their help during wheat harvest. Further, we are grateful to Florian Enz and Sophie Gulliver for field and laboratory assistance. This work was supported by the Interfaculty Research Collaboration "One Health" of the University of Bern.

## Additional information

### Funding

| Funder | Grant reference number | Author |
|---|---|---|
| University of Bern | Interfaculty Research Collaboration "One Health" | Klaus Schlaeppi Matthias Erb |

The funders had no role in study design, data collection and interpretation, or the decision to submit the work for publication.

### Author contributions

Valentin Gfeller, Conceptualization, Data curation, Formal analysis, Investigation, Visualization, Methodology, Writing - original draft, Writing - review and editing; Jan Waelchli, Data curation, Formal analysis, Visualization; Stephanie Pfister, Data curation, Investigation; Gabriel Deslandes-Hérold, Investigation, Methodology; Fabio Mascher, Yvo Aeby, Resources, Investigation; Gaetan Glauser, Resources, Data curation, Investigation; Adrien Mestrot, Christelle AM Robert, Resources, Writing - review and editing; Klaus Schlaeppi, Conceptualization, Data curation, Formal analysis, Supervision, Investigation, Methodology, Writing - original draft, Writing - review and editing; Matthias Erb, Conceptualization, Resources, Formal analysis, Supervision, Funding acquisition, Investigation, Methodology, Writing - original draft, Writing - review and editing

### Author ORCIDs

Valentin Gfeller ![ORCID] http://orcid.org/0000-0001-8896-7280
Gabriel Deslandes-Hérold ![ORCID] http://orcid.org/0000-0002-5072-1450
Gaetan Glauser ![ORCID] http://orcid.org/0000-0002-0983-8614
Christelle AM Robert ![ORCID] http://orcid.org/0000-0003-3415-2371
Matthias Erb ![ORCID] http://orcid.org/0000-0002-4446-9834

### Decision letter and Author response

Decision letter https://doi.org/10.7554/eLife.84988.sa1
Author response https://doi.org/10.7554/eLife.84988.sa2

## Additional files

### Supplementary files

• Supplementary file 1. Specifications to library preparation for sequencing.
• Supplementary file 2. Barcode-to-sample assignments and meta data of microbiome analysis.
• MDAR checklist

### Data availability

All data related to this study is shown in form of figures and tables in the manuscript. The raw sequencing data is available from the European Nucleotide Archive (http://www.ebi.ac.uk/ena) with the study

accession PRJEB53704 and the sample IDs SAMEA110170660 (BE09) and SAMEA110170661 (BE10). All other raw data is included as source data files. The code to reproduce the bioinformatic pipeline and statistical analysis can be downloaded from GitHub (https://github.com/PMI-Basel/Gfeller_et_al_Posieux_field_experiment, copy archived at *Gfeller, 2023*).

The following dataset was generated:

| Author(s) | Year | Dataset title | Dataset URL | Database and Identifier |
|---|---|---|---|---|
| Schlaeppi K | 2023 | Plant secondary metabolite-dependent plant-soil feedbacks can improve crop yield in the field | http://www.ebi.ac.uk/ena/data/view/PRJEB53704 | European Nucleotide Archive, PRJEB53704 |

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
