## [Editor Report]

This study presents findings that are important for understanding plant-soil feedbacks in agriculture. The authors use a large-scale agricultural field experiment to demonstrate the role of root-emitted secondary metabolites in enhancing the yield of the next crop. By using a benzoxazinoid-deficient maize genotype, the authors provide compelling evidence that biomass production and grain yield of several wheat varieties can be increased when grown in soil conditioned by maize plants able to release benzoxazinoids.

---

## [Decision Letter]

**Decision letter after peer review:**

Thank you for submitting your article "Plant secondary metabolite-dependent plant-soil feedbacks can improve crop yield in the field" for consideration by *eLife*. Your article has been reviewed by 2 peer reviewers, and the evaluation has been overseen by a Reviewing Editor and Meredith Schuman as the Senior Editor. The following individuals involved in the review of your submission have agreed to reveal their identity: Benjamin Delory (Reviewer #1); Ian Kaplan (Reviewer #2).

Essential revisions:

1) Please consider the detailed comments provided by the reviewers for clarifying several parts of the material and methods and elsewhere in the text.

2) Please, take into consideration re-analysing part of your data to consider the specifics of your experimental design. From either mine or the reviewers' understanding, you have here a split-plot design, which would thus require taking at least into account the strips of your field experiment as a random blocking factor in the analysis, to avoid potential pseudo-replication.

Reviewer #2 (Recommendations for the authors):

L24-25. I don't know what these two sentences mean ("Weed cover…across varieties"). In comparison to what? It's unclear as written.

L39. It's a little unclear what you mean by "mechanistic work" in this context.

Also, it could be useful, either here or elsewhere, to differentiate proximate vs. ultimate mechanisms. This study is aiming at proximate mechanisms, but there has been work on ultimate mechanisms underlying plant-soil feedbacks in crop rotation. See, for example:

I Kaplan, NA Bokulich, JG Caporaso, LS Enders, W Ghanem, … et al. 2020. Phylogenetic farming: Can evolutionary history predict crop rotation via the soil microbiome?

Evolutionary Applications 13 (8), 1984-1999

L48. This is an awkward expression: "genetically defined conditions".

L70. Positive or negative feedbacks?

L90. Can you clarify what a "strong reduction" means in more quantitative terms? Does bx1 produce zero? Or only trace amounts? Compared to the wild-type what sort of overall reduction is occurring?

L93-94. I understand after looking at Figure S1, but this is not described very well. I don't think people will know what this means: "Wild type and bx1 mutant plants were alternatingly sown in 5 strips". It was 10 strips, not 5 strips. It's not intuitive that you're saying that you had 5 strips of each variety (10 total) that were alternated with one another spatially across the field. Try to reword for clarity.

L106-109. Can you explain this better? I still don't follow why you found the reverse effect here.

L116-117. I don't know what you mean by "for multiplication" and "by a third party". Try to reword for clarity.

L143. Unclear what you mean by "compartments". I'm guessing this is talking about soil vs. rhizosphere vs. root, but not obvious.

L212. It looks like you're organizing the sections chronologically through the growth cycle, but it might make more sense to bump back the wheat emergence/growth section so it's combined here. In my mind, all the plant performance data, early- and late-season, are closely aligned and having weeds/insects split it up disrupts the flow a bit.

L261. Can you comment somewhere in here on how your wheat yields compare with commercially reported yields in Switzerland? Those data must be known and easily accessible. It would just be helpful to know what your wheat yield increases mean compared to grower fields.

L288-289. Interesting! I like this comparison of magnitude with breeding. This is a helpful perspective.

L309. "live" should be singular – "life".

L331. Missing a word in this line – "and" are commonly cultivated?

L342. "phenotype a subset of phenotypes" – is this wording correct?

L344. Insert a comma after "phase".

L366. Insert a comma after "(n=10)".

L464. Change “leave” to “leaf”.

Figure 2A vs. 2C. I wouldn't use different terms to mean the same thing, it gets too confusing. For example, "run" is the same in tables A and C, but you switch from "genotype" to "conditioning" to mean maize genotype. This is especially confusing since you also include "variety" in C, which can get easily confused with genotype.

Figures3, 5, and 6. I'm not a fan of having the y-axis min value something greater than zero. Yes, it's better for visualizing differences but can be highly misleading of the magnitude of treatment effects if the y min doesn't start at zero.

Figure S1 – this is a very helpful figure for understanding your experimental design!

---

## [Author Response]

Essential revisions:1) Please consider the detailed comments provided by the reviewers for clarifying several parts of the material and methods and elsewhere in the text.

We are grateful for the constructive comments of the two reviewers and did our best to integrate them into the revised version of the manuscript. Please find our response below.

2) Please, take into consideration re-analysing part of your data to consider the specifics of your experimental design. From either mine or the reviewers' understanding, you have here a split-plot design, which would thus require taking at least into account the strips of your field experiment as a random blocking factor in the analysis, to avoid potential pseudo-replication.

Thanks for pointing out how we can improve our statistical analysis. We reanalyzed our data and updated the figures and the main text of the manuscript accordingly. The analysis was modified as described in the methods section:

“To account for our experimental design and avoid potential pseudo-replication, we included the maize strips as a random blocking factor. To accomplish this, we fitted a linear mixed-effects model using the *lme()* function from the *nlme* package (Pinheiro et al., 2021). This function further allowed us to account for unequal variance among treatment groups, if necessary.”

Conclusions remain unaffected, with the exception of the plant damage effect, which moved from a “statistically significant” effect in one variety to a tendency. The manuscript was adjusted accordingly.

Reviewer #2 (Recommendations for the authors):L24-25. I don't know what these two sentences mean ("Weed cover…across varieties"). In comparison to what? It's unclear as written.

Thanks for pointing this out. The two sentences refer to comparisons between soil conditioning treatments as described in the previous sentence. We have now clarified this aspect.

L39. It's a little unclear what you mean by "mechanistic work" in this context.Also, it could be useful, either here or elsewhere, to differentiate proximate vs. ultimate mechanisms. This study is aiming at proximate mechanisms, but there has been work on ultimate mechanisms underlying plant-soil feedbacks in crop rotation. See, for example:I Kaplan, NA Bokulich, JG Caporaso, LS Enders, W Ghanem, … et al. 2020. Phylogenetic farming: Can evolutionary history predict crop rotation via the soil microbiome?Evolutionary Applications 13 (8), 1984-1999

Thank you. We now specify that we refer to proximate mechanisms.

“So far however, proximate mechanistic work on plant-soil feedbacks has rarely been applied to improve crop rotations.”

L48. This is an awkward expression: "genetically defined conditions".

We have rewritten this sentence.

“The response to plant-soil feedbacks can strongly depend on environmental conditions (Smith‐Ramesh and Reynolds, 2017), and is often species- and variety-specific, thus requiring detailed investigations of defined plant genotypes under realistic environmental conditions (van der Putten et al., 2013; Wagg et al., 2015; Hu et al., 2018b; Cadot et al., 2021a).”

L70. Positive or negative feedbacks?

We now provide more information.

“Soil conditioning by benzoxazinoids can feed back on growth and defense of maize and wheat, where the strength and direction of the feedback depend on the plant genotype and soil characteristics (Hu et al., 2018b; Cadot et al., 2021a).”

L90. Can you clarify what a "strong reduction" means in more quantitative terms? Does bx1 produce zero? Or only trace amounts? Compared to the wild-type what sort of overall reduction is occurring?

Given the limits of detection, the reduction of the highly emitted compounds (e.g. HDMBOA-Glc) is at least 100-fold. We included this information in the revised manuscript.

“Most benzoxazinoids were below the limit of detection in the soils planted with *bx1* mutant plants, indicating that concentrations of the highly emitted compounds (e.g. HDMBOA-Glc) were more than 100 times lower in *bx1* mutant compared to wild type conditioned soil.”

L93-94. I understand after looking at Figure S1, but this is not described very well. I don't think people will know what this means: "Wild type and bx1 mutant plants were alternatingly sown in 5 strips". It was 10 strips, not 5 strips. It's not intuitive that you're saying that you had 5 strips of each variety (10 total) that were alternated with one another spatially across the field. Try to reword for clarity.

We did reword this part for clarity.

“Wild type and *bx1* mutant plants were sown separately in 10 strips. The strips themselves were arranged in an alternating pattern, with a strip containing wild type plants followed by a strip containing *bx1* mutant plants, and so on (Figure S1). Each strip consisted of 12 rows of maize of one genotype.”

L106-109. Can you explain this better? I still don't follow why you found the reverse effect here.

We reformulated this sentence to clarify.

“Only trace levels of the two glycosylated benzoxazinoids, DIMBOA-Glc and HDMBOA-Glc were found, and their concentrations were higher in *bx1* mutant conditioned soils. As benzoxazinoids are released as glycosides and deglycosylated in the soil, this result is indicative of faster deglycosylation in wild type conditioned soils.”

We also included the following to clarify why we found benzoxazinoids in *bx1* mutant conditioned soil.

“Most benzoxazinoids were 3- to 800-fold less abundant than at the end of maize cultivation (Figure 1B). We therefore concentrated the samples prior to analysis, resulting in the detection of traces of benzoxazinoids also in *bx1* mutant conditioned soils.”

L116-117. I don't know what you mean by "for multiplication" and "by a third party". Try to reword for clarity.

We have rewritten this sentence.

“An additional variety (Sailor) was sown for seed multiplication adjacent to the two other varieties on the same field by a seed company.”

L143. Unclear what you mean by "compartments". I'm guessing this is talking about soil vs. rhizosphere vs. root, but not obvious.

We specified what we mean by compartments.

“At maize harvest, the biggest taxonomic differences at the phylum level were found between the three compartments: root, rhizosphere, and soil (Figure S3).”

L212. It looks like you're organizing the sections chronologically through the growth cycle, but it might make more sense to bump back the wheat emergence/growth section so it's combined here. In my mind, all the plant performance data, early- and late-season, are closely aligned and having weeds/insects split it up disrupts the flow a bit.

Thanks for this input. After discussion among authors, we have decided to maintain the chronological order.

L261. Can you comment somewhere in here on how your wheat yields compare with commercially reported yields in Switzerland? Those data must be known and easily accessible. It would just be helpful to know what your wheat yield increases mean compared to grower fields.

Thank you. We added this information.

“Yield in both Claro and Fiorina was increased by 4-5 % on benzoxazinoid conditioned soils (Figure 6A); productivity of both varieties fell within the expected range of top-yielding winter wheat varieties in Switzerland (variety testing Agroscope).”

L288-289. Interesting! I like this comparison of magnitude with breeding. This is a helpful perspective.L309. "live" should be singular – "life".

Done.

L331. Missing a word in this line – "and" are commonly cultivated?

Done.

L342. "phenotype a subset of phenotypes" – is this wording correct?

Done.

L344. Insert a comma after "phase".

Done.

L366. Insert a comma after "(n=10)".

Done.

L464. Change “leave” to “leaf”.

Done.

Figure 2A vs. 2C. I wouldn't use different terms to mean the same thing, it gets too confusing. For example, "run" is the same in tables A and C, but you switch from "genotype" to "conditioning" to mean maize genotype. This is especially confusing since you also include "variety" in C, which can get easily confused with genotype.

Thank you for this input. Clarified by writing “wheat variety” in 2C. We kept the wording “genotype” and “conditioning”, because we consistently used these two notions for the conditioning phase and the response phase.

Figures3, 5, and 6. I'm not a fan of having the y-axis min value something greater than zero. Yes, it's better for visualizing differences but can be highly misleading of the magnitude of treatment effects if the y min doesn't start at zero.

We consider this acceptable for boxplots. To make the reader aware of our choice we indicated our procedure in figure captions.

“Note that the minimum values of the y-axes were set to a value greater than zero for clearer visualization of treatment differences.”

Figure S1 – this is a very helpful figure for understanding your experimental design!

Many thanks for this feedback.